# High crosstalk suppression in InGaAs/InP single-photon avalanche diode arrays by carrier extraction structure

Yongsheng Tang ®[1,2], Rui Wang[1], Xiaohong Yang ®[1,3] ✉, Tingting He[1,3], Yijun Liu[1,3] & Meng Zhao[1]

Crosstalk has become an urgent issue for single-photon avalanche diode arrays. In previous work, trenches were introduced between pixels to block the crosstalk optical path in planar InGaAs/InP single-photon avalanche diode arrays, since the optical crosstalk was considered as the main crosstalk mechanism. However, the crosstalk suppression effect of this solution is not satisfactory. Here, we demonstrate a carrier extraction structure to efficiently reduce crosstalk by electrically guiding photogenerated crosstalk holes in the non-pixel region to the surface, since we find that the optical-electrical crosstalk is the dominant crosstalk mechanism. Experimental measurements show that a narrow carrier extraction structure makes a 91.52% (96.22%) crosstalk reduction between the nearest neighbor pixels in arrays with 100 (50) μm pixel pitch, and it does not cause any etching damage. These results reveal the primary source of crosstalk in InGaAs/InP single-photon avalanche diode arrays and provide a practical route to fabricate low-crosstalk, high-pixel-density arrays for use in high-resolution three-dimensional imaging and quantum technologies.

Single-photon avalanche diodes (SPADs) and arrays have been widely used in quantum communication, three-dimensional light detection and ranging (3-D LiDAR), and medical imaging[1–5], benefiting from their high detection sensitivity of a single photon, sub-nanosecond timing resolution, high reliability, low cost, and compactness[6]. In recent years, single-photon LiDAR based on SPADs and arrays has been advanced and enabled long-range 3-D imaging from tens to hundreds of kilometers[7–9]. Among them, InGaAs/InP SPADs and arrays are the most promising detectors for eye-safe short-wavelength infrared range (SWIR) applications due to their good performance. However, the drawback of crosstalk[10,11] between pixels needs to be solved to obtain high-quality signals in SPAD arrays. Crosstalk is an unwanted count triggered by another avalanche event in the array, which would dete-riorate the signal-to-noise ratio (SNR) and reduce the spatial resolution of the imaging[11–13]. This effect is caused by the crosstalk photon emission in the InP multiplication region: when an avalanche event is triggered within a pixel, large avalanche hot carriers flow through the junction; simultaneously, near-infrared crosstalk photons in the wavelength range of 900–1700 nm[12] are emitted owing to hot carriers recombination[12,14,15]. These crosstalk photons can be detected by adjacent pixels, leading to crosstalk. The crosstalk intensity increases as the pixel pitch decreases, thereby restricting the attainable pixel density of SPAD arrays.

In previous work, the optical crosstalk was considered the primary crosstalk mechanism in InGaAs/InP SPAD arrays, where crosstalk photons are coupled to the active area of adjacent pixels via direct and indirect optical paths[10–12,16–22]. For mesa devices, a crosstalk filter layer and backside metallization have been employed to attenuate crosstalk photons coupled through the indirect optical path, and the crosstalk probability was reduced by about one order of magnitude[12]. However,

[1]State Key Laboratory of Integrated Optoelectronics, Institute of Semiconductors, Chinese Academy of Sciences, Beijing, China. [2]School of Electronic, Electrical and Communication Engineering, University of Chinese Academy of Sciences, Beijing, China. [3]College of Materials Science and Opto-Electronic Technology, University of Chinese Academy of Sciences, Beijing, China. ✉e-mail: xhyang@semi.ac.cn

the exposed sidewalls of the multiplication region and etching damage would increase the dark current, which results in high dark count rates (DCR) and low reliability. Therefore, the planar device is preferred due to its internal multiplication region structure, low dark current (<1 nA), and high reliability. For planar devices, previous efforts to suppress crosstalk have focused on blocking the direct optical path through etching trenches between SPADs[16–18,22–24]. The crosstalk suppression between the nearest neighbor pixels is ~60% by using Focused Ion Beam (FIB) etching and filling trenches with Platinum[22], but this solution has limited crosstalk suppression. The main explanation for this limitation was the contribution of the indirect optical crosstalk. In fact, we find that the contribution of the optical crosstalk to the total crosstalk is small. Most crosstalk photons are absorbed in non-pixel regions, and the pixel active area has a high collection efficiency for these photogenerated holes. Apart from optical crosstalk, there is also optical-electrical crosstalk[25] in planar InGaAs/InP SPAD arrays, which dominate the crosstalk. The optical-electrical crosstalk is that crosstalk photons generated in the *emitter* pixel travel toward adjacent pixels and are absorbed in the InGaAs layer; then, these photogenerated holes diffuse to adjacent pixels and are captured by the depletion region. However, the influence and suppression of this optical-electrical crosstalk have hardly been investigated.

In this paper, we demonstrate a carrier extraction structure (CES) in planar InGaAs/InP SPAD arrays that realize high crosstalk suppression. This CES is p-doped channels to isolate pixels by guiding photogenerated crosstalk holes to the surface, rather than blocking optical paths. We have analyzed the feasibility of this CES in terms of the most fundamental carrier transport property of the device and verified it in experiments, results show that the crosstalk suppression effect exceeds 90% in arrays with 100/50 μm pixel pitch due to the strong suppression of the optical-electrical crosstalk. Better crosstalk suppression can be achieved by reducing the distance between the pixel and the CES. When the distance is larger than 12 μm, efficient crosstalk suppression can be achieved even if a 0 V bias voltage is applied on the CES. As the distance decreases, a higher reverse bias voltage on the CES is required to guarantee pixels and CES work independently.

Additionally, arrays with higher pixel density can be fabricated by adopting a narrow CES closer to the pixels while achieving high crosstalk suppression. Our results reveal that the optical-electrical crosstalk is the dominant crosstalk mechanism in planar InGaAs/InP SPAD arrays, and importantly, these results provide an approach to facilitate the development and application of low-crosstalk, high-pixel-density SPAD arrays. It should be noted that this technique can also be used in other SPAD arrays, but works better in arrays with different absorption and multiplication materials.

## Results

### InGaAs/InP SPAD structure

The typical separate absorption, grading, charge, and multiplication (SAGCM) heterostructure[26–28] was adopted in investigated InGaAs/InP SPADs. The photon absorption layer is an intrinsic $In_{0.53}Ga_{0.47}As$ with a thickness of 2 μm, an electron–hole pair is generated after the photon is absorbed. The photogenerated hole separated by electric field injects into the high-field InP multiplication layer and triggers a self-sustaining avalanche multiplication, thus generating a macroscopic current pulse that can be easily detected by the external circuit. (More details about the device structure and fabrication processes can be found in Methods or our previous work[29]).

### Design of carrier extraction structure (CES)

This CES is a p-doped channel to isolate pixels and guide crosstalk photogenerated holes to the surface to suppress the crosstalk. Figure 1 shows the two-dimensional (2-D) cross-section schematic of the reported planar InGaAs/InP SPAD arrays with the carrier extraction structure (CES). The CES is formed via a zinc (Zn) deep diffusion in a Metal-Organic Chemical Vapor Deposition (MOCVD) reactor. The diffusion depth is the total thickness of the InP cap layer and the InP charge layer, which is 3.7 μm in this paper. Additionally, the width of the Zn diffusion window is $D$. While the actual width of CES is greater than $D$ due to the lateral diffusion, the ratio of the lateral diffusion length to the vertical diffusion depth is ~0.7. (The p-InP diffusion region contours of the pixel and the CES can be found in

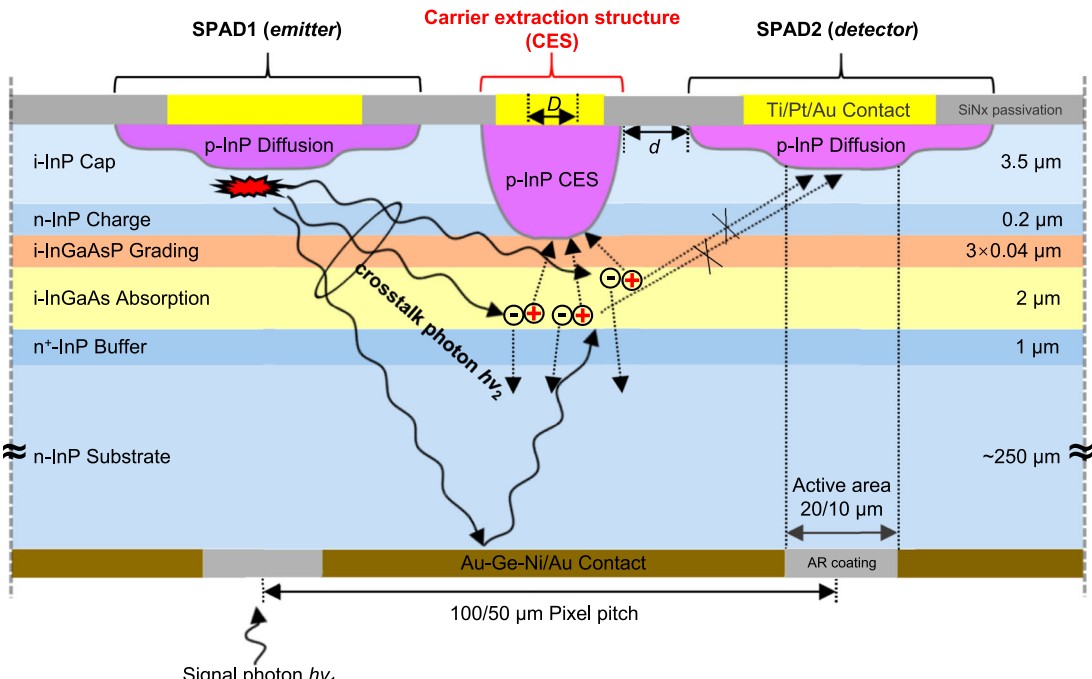

**Fig. 1 | Planar InGaAs/InP single-photon avalanche diode (SPAD) array with the carrier extraction structure (CES).** Two-dimensional (2-D) cross-section schematic of the nearest neighbor pixels. $D$ is the width of the Zn diffusion window for the CES, and $d$ is the distance between the pixel and the CES. $hv_1$ represents the signal photon, and $hv_2$ represents the crosstalk photons emitted during SPAD1 self-sustaining avalanche multiplication.

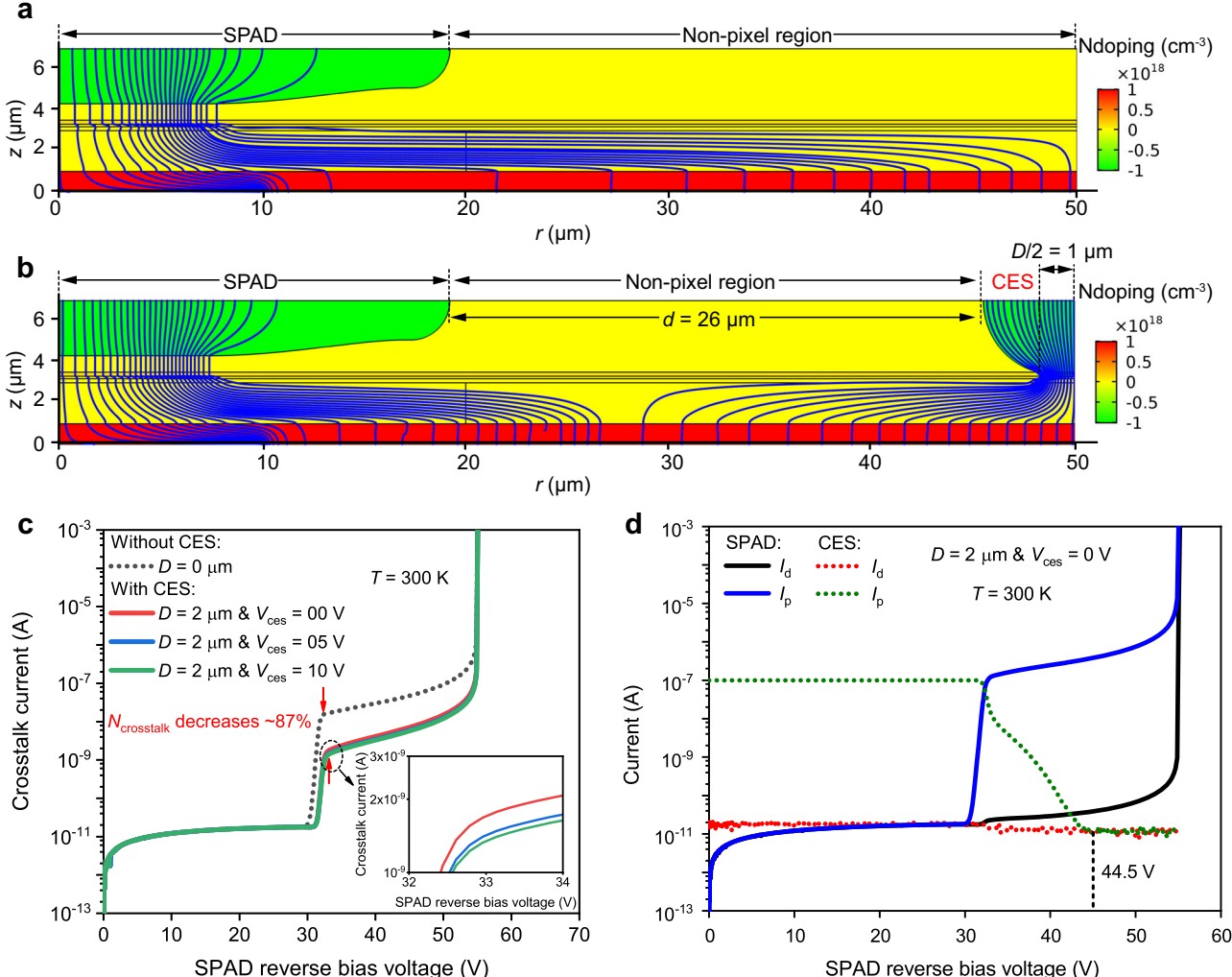

**Fig. 2 | Simulation of the crosstalk suppression effect at a temperature (*T*) of 300 K.** The distribution of crosstalk current at unity gain (single-photon avalanche diode (SPAD) reverse bias voltage = punch-through voltage, $V_{ph}$) for **a** the conventional device (without carrier extraction structure (CES)), and **b** the device with CES, the reverse bias voltage applied on the CES ($V_{ces}$) is 0 V. Figures show the doping concentration in each layer (color legend on the right) and the crosstalk current density (blue streamlines). The CES diffusion window width (*D*) is set to 2 μm. **c** Simulated crosstalk current-voltage (*I*–*V*) curves of the SPAD. Inset: the enlarged portion near $V_{ph}$, illustrating when the CES is operated at a larger $V_{ces}$, the crosstalk suppression is slightly improved. **d** Simulated dark current ($I_d$) and photocurrent ($I_p$) voltage characteristics of the SPAD (solid lines) and the CES (dotted lines).

Supplementary Fig. 3). The distance between the pixel and the CES is *d*. In addition, the reverse bias voltage $V_{ces}$ between this p-doped channel and n-contact is applied to guide crosstalk carriers for crosstalk suppression, rather than blocking their paths by a trench. This CES is equivalent to forming a PIN detector in non-pixel regions to collect crosstalk carriers.

## CES crosstalk suppression effect simulation

To evaluate the crosstalk suppression capability of our proposed CES, we developed a 2-D rotationally symmetric model in COMSOL Multiphysics to simulate crosstalk current-voltage (*I*–*V*) characteristics of the pixel. The structures of model are displayed in Fig. 2a, b, where Fig. 2a shows the pixel without CES, and Fig. 2b shows the pixel with CES, with a *D* value of 2 μm and a *d* value of 26 μm. A crosstalk carrier generation rate was defined in the whole InGaAs absorption layer to simulate the absorption of crosstalk photons. Here both the conventional optical crosstalk and the optical-electrical crosstalk were considered, so the simulation results show the total crosstalk and crosstalk suppression. (To show the crosstalk suppression more clearly, the crosstalk carrier generation rate defined is amplified, so the crosstalk current in simulated results is much larger than the real one. And the

attenuation of crosstalk photons with the propagation distance has also been considered.) The crosstalk and crosstalk suppression were mainly characterized by DC crosstalk measurements based on SPAD's DC *I*–*V* characteristics; they were quantified by the number of carriers ($N_{crosstalk}$) captured at the unity gain, which is the intrinsic source of avalanche counts. (More details about DC crosstalk measurements can be found in Methods.) The distribution of the crosstalk current density at unity gain is shown in Fig. 2a, b, where the blue streamlines represent the crosstalk current density, and the color legend on the right represents the doping concentration of each layer. For the conventional device (without CES), the crosstalk photogenerated holes are entirely collected by the active area, as shown in Fig. 2a. However, for the device with CES, the majority of the crosstalk photogenerated holes are collected by the CES, which greatly suppresses crosstalk, as shown in Fig. 2b.

The crosstalk *I*–*V* characteristics were also recorded, as shown in Fig. 2c. The crosstalk current ($I_v$) increases significantly near punch-through voltage ($V_{ph}$, 32.5 V) and has an obvious and smooth gain with increasing SPAD reverse bias voltage, indicating that these crosstalk carriers collected by the SPAD can provide avalanche gain and cause crosstalk. Compared to the device without CES, when the CES is biased

at 0 V ($V_{ces} = 0$ V), $I_v$ decreases from 16.56 nA to 2.02 nA at unity gain, as the $N_{crosstalk}$ is reduced from $1.04 \times 10^{11}$ cps to $1.26 \times 10^{10}$ cps, showing 87.88% crosstalk suppression. In addition, increasing $V_{ces}$ results in better crosstalk suppression, due to a slight increase in the lateral electric field in the InGaAs layer. At $V_{ces} = 10$ V, $I_v$ further decreases to 1.64 nA, and the crosstalk suppression effect is improved to 90.10%, as shown in the inset in Fig. 2c.

Furthermore, the simulation results also demonstrate that the introduction of the CES would not reduce the photon detection efficiency (PDE) of the pixel. A carrier generation rate (uniform electron–hole pair generation rate ($1e27\,\mathrm{m^{-3}\,s^{-1}}$)) was defined in the active area to simulate the absorption of signal photons. Figure 2d shows the simulated dark current ($I_d$) and photocurrent ($I_p$) voltage characteristics of the SPAD and CES, where solid lines represent the current of SPAD and dotted lines represent the current of CES. The reverse bias voltage of the SPAD varies from 0 V to breakdown voltage ($V_{br}$), while $V_{ces}$ is held constant at 0 V. It is shown that the $I_d$ and $I_p$ of the CES remain consistent when the SPAD reverse bias voltage is above 44.5 V, indicating that all the photogenerated holes in the active area inject into the InP multiplication layer. On the other hand, to reduce the contribution of the thermal generation to the dark count, the operating temperature of InGaAs/InP SPADs is usually cooled to ~230 K[18,30,31]. The $V_{br}$ and the operating temperature show a linear positive dependence, 118 mV/K[32] for our device. The $V_{br}$ of our SPADs at 300 K is -55 V; therefore, when they are operated in Geiger mode (GM) at 230 K, the operating voltage remains above 44.5 V and the CES does not reduce the pixel PDE. Hence, simulation results have proven that this CES can realize high crosstalk suppression without degrading the optical response of pixels.

## Measurement results and analysis

We fabricated arrays with the same structure as the simulation model, and measured results of them show that the CES achieves more than 90% crosstalk suppression. Crosstalk characteristics were investigated between two nearest neighbor pixels. Figure 3a shows the anode surface micrographs of the nearest neighbor pixels for the fabricated 64 × 64 planar InGaAs/InP SPAD arrays without (left) and with (middle and right) CES. Crosstalk intensifies as the pixel density increases. Therefore, to investigate the crosstalk suppression of this CES for arrays with different pixel densities, two arrays with different pixel pitches were fabricated from the same wafer. In the array with a 100 (50) μm pixel pitch, the active area diameter is 20 (10) μm, the $D$ value is set to 2 μm, and the $d$ is -26 (6) μm. The DC crosstalk measurements were mainly adopted to evaluate crosstalk characteristics, one pixel acted as the *emitter* (crosstalk source), which was biased above the breakdown at a constant current to continuously emit crosstalk photons. Another pixel was used as the *detector* (victim pixel) to characterize the crosstalk, the $I_v$ and $N_{crosstalk}$ at unity gain obtained from the measured DC $I–V$ data when the *emitter* was ON and OFF. The $I_d$ of the *detector* was measured when the *emitter* was OFF, and the $I_v$ was measured when the *emitter* was kept ON. (More detailed test setups and processes can be found in the Methods section and Supplementary Fig. 7).

The crosstalk characteristics between the nearest neighbor pixels in arrays with 100 μm pixel pitch were investigated at first. Figure 3b shows the measured $I$-$V$ curves of the *detector* at temperature of 300 K, where solid and dotted lines represent the currents in the array without and with the CES, respectively. $I_d$ and $I_v$ are almost the same before $V_{ph}$, but $I_v$ increases significantly at unity gain. $I_{v1}$ increases from 0.07 nA to 0.68 nA, and $I_{v2}$ increases from 0.07 nA to 0.13 nA, which are mainly caused by the absorption of long-wavelength crosstalk photons in the InGaAs layer. Therefore, the long-wavelength crosstalk photons dominate the crosstalk in InGaAs/InP SPAD arrays. The large increment in $I_v$ is due to the fact that the *emitter* continues to work at a constant current of 300 μA, which amplifies the actual crosstalk current. In addition, the crosstalk-current gain ($M_{v1}$, $M_{v2}$) rises smoothly, $M_{v1}$ and $M_{v2}$ increase to 98 and 63 before the breakdown. This is similar to the photocurrent gain when the light is absorbed in the active area, suggesting that these crosstalk carriers collected by the *detector* could provide GM avalanche gain, leading to crosstalk counts.

The CES shows high crosstalk suppression thanks to the strong suppression of the optical-electrical crosstalk. As shown in Fig. 3c, at a working current of 200 μA for the *emitter*, the crosstalk of the array with CES is reduced by 90.47% compared to the array without CES, a narrow CES can realize high crosstalk suppression. Since the CES does not degrade the optical response of the pixel, or the CES mainly collects crosstalk carriers in non-pixel regions. These results reveal that the conventional optical crosstalk is not the main crosstalk mechanism in planar InGaAs/InP SPAD arrays, but the optical-electrical crosstalk. This is different from Si arrays, the main reason is that the InGaAs/InP SPADs have different absorption and multiplication materials and there is a larger cut-off wavelength in the narrow band absorption layer. It is also the main reason why the crosstalk probability of InGaAs/InP arrays is much larger than Si arrays[13,22].

Moreover, crosstalk suppression increases from 90.47% to 91.52% as the $V_{ces}$ increases from 0 V to 10 V, as shown in Fig. 3c. However, the $V_{ces}$ cannot be too large to make any avalanche in the InGaAs layer. Figure 3c also shows the linearly increasing relationship between the $N_{crosstalk}$ and the *emitter* working current. This is because the number of emitted crosstalk photons is proportional to the total avalanche charge $Q$ flowing through the junction. Therefore, we can also reduce the $Q$ by reducing the capacitance $C$, applying a lower excess bias voltage $V_{ex}$ (In the passive quenching, the total avalanche charge $Q$ can be expressed as $Q \cong C \times V_{ex}$[33]), or using fast active quenching circuits[19] to further suppress crosstalk.

We also investigated the crosstalk suppression of the CES when the *detector* was operated in Geiger mode, the CES performance is basically the same as the DC crosstalk measurements with a little difference. The results are shown in Fig. 3d, at a working current of 200 μA for the *emitter* and an excess bias voltage of 0.5 V for the *detector* ($V_{ex\_D} = 0.5$ V), the average crosstalk suppression is 88.60% at $V_{ces} = 0$ V, and 89.82% at $V_{ces} = 10$ V.

Experimental results also show that for arrays with CES, the array with a pixel pitch of 50 μm has better crosstalk suppression than the array with a pixel pitch of 100 μm. The measured $I$-$V$ characteristics are shown in Fig. 3f. A crosstalk reduction of 95.46% is observed at $V_{ces} = 10$ V. And as the $V_{ces}$ increases to 30 V, the crosstalk suppression is improved to 96.22%. These low-crosstalk SPAD arrays can be applied for high-resolution 3-D imaging. It is important to note that, however, for the array with a 50 μm pixel pitch, the value of $d$ is -6 μm, as shown in Fig. 3a (right). So, a $V_{ces}$ above 10 V is required to guarantee the pixel and the CES work independently, as the measured results of the unit pixel shown in Fig. 3e.

## Physical analysis and bias condition optimization for the CES

To obtain better crosstalk suppression effect and discuss the application of the CES in the array with higher pixel density, we investigated $I–V$ characteristics of the pixel at different distances $d$. The simulated DC $I–V$ characteristics are shown in Fig. 4a, where the distance $d$ between the pixel and CES varied from 26 μm to 7 μm while $V_{ces}$ remained at 0 V. When the value of $d$ ranges from 26 μm to 17 μm, the pixel works normally. However, as $d$ decreases, an extra current component contributes to the pixel current at the pixel bias voltage of -22 V (1 V) when $d = 12$ (7) μm. This current component would be dominant as the $d$ value shrinks to 7 μm, which may cause pixel failure. Further simulated results indicate that this abnormal current component is the hole current that flows from the CES to the pixel anode, as shown in the pink dotted square box in Fig. 4b. Figure 4b is the flow chart of the current density at 90% breakdown voltage ($0.9V_{br}$, 49.5 V),

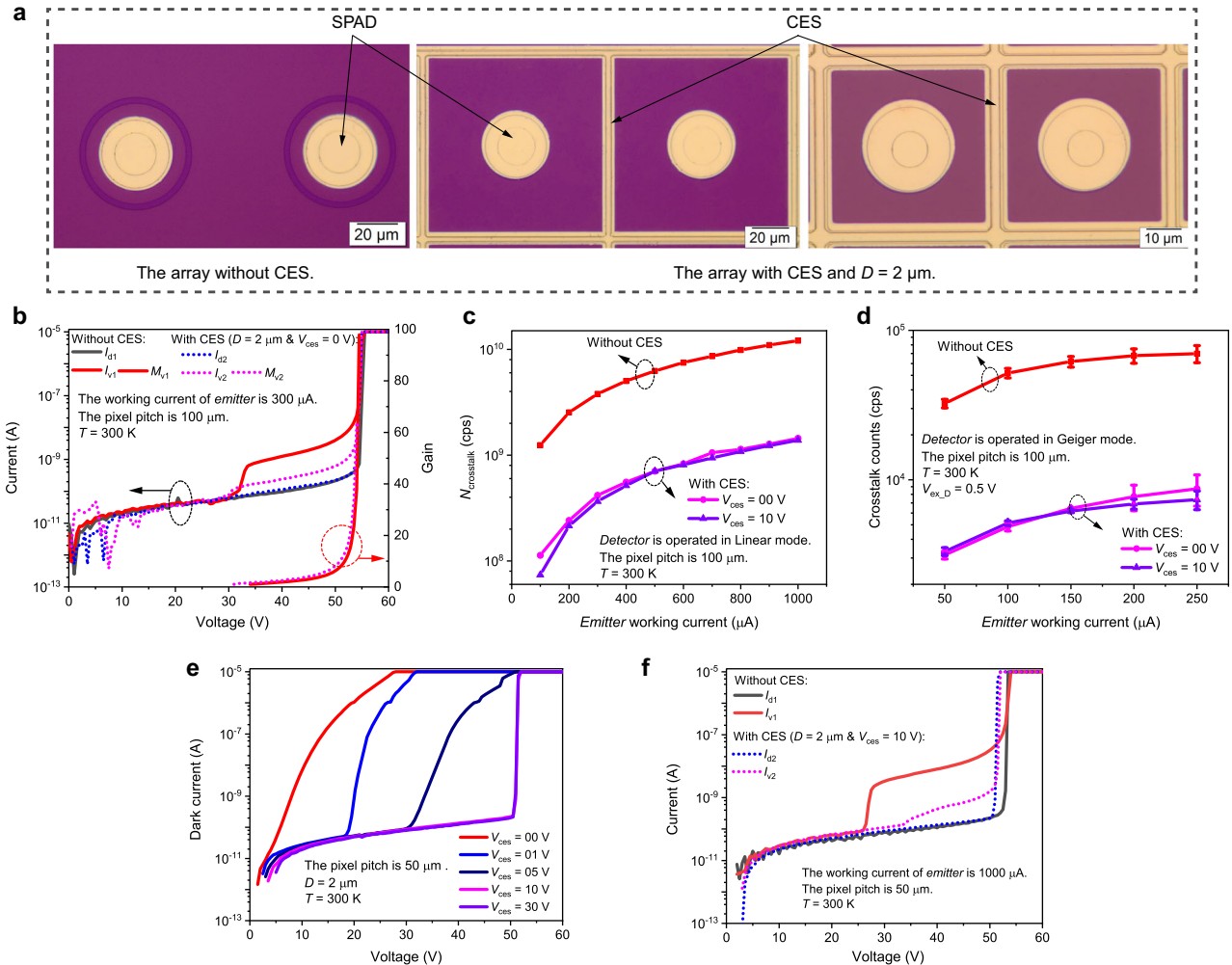

**Fig. 3 | The measured crosstalk suppression of the carrier extraction structure (CES) between the nearest neighbor pixels at a temperature (T) of 300 K. a** Top view of the nearest neighbor pixels in the array has a 100 μm pixel pitch but without CES (left), has a 100 μm pixel pitch and with CES (middle), has a 50 μm pixel pitch and with CES (right). The CES diffusion window width (D) is 2 μm. **b–d** Measured crosstalk suppression of the CES in the array with 100 μm pixel pitch. **b** Measured current-voltage (I-V) characteristics and gain voltage (M-V) characteristics of the *detector*, when the *emitter* avalanched at a constant current of 300 μA and the CES is biased at 0 V ($V_{ces}$ = 0 V). $I_{v1}$ (solid line) and $I_{v2}$ (dotted line) represent the crosstalk current of the array without and with CES, respectively. **c** The relationship between

the number of carriers collected by the *detector* ($N_{crosstalk}$) and the *emitter* working current, when the *detector* was operated in Linear mode. **d** The relationship between the crosstalk counts and the *emitter* working current, when the *detector* was operated at 0.5 V excess voltage ($V_{ex\_D}$). The error bar represents the fluctuation in counts between repeated tests. **e-f** Measured results in the array with 50 μm pixel pitch. **e** The unit pixel's dark I-V characteristics when the $V_{ces}$ is from 0 V to 30 V. A $V_{ces}$ exceeding 10 V is required to ensure the pixel and the CES work independently because of the close spacing between them. **f** Measured I-V characteristics of the *detector* in arrays without (solid lines) and with (dotted lines) CES, when the *emitter* avalanched at 1000 μA constant current.

when $d$ = 12 μm. To enhance visibility, we display limited simulation areas of 0–40 μm, rather than the complete 0–50 μm simulation regions. The cyan streamlines represent the current density, and the color legend on the right represents the electric field. This hole leakage current is mainly distributed in a thin layer near the surface.

To investigate the origin of this hole current and block it to ensure that pixels and CESs work independently, we studied the distribution of the valence band between the pixel and the CES. Figure 4c shows the valence band distribution in the device cross-section when the pixel is biased at $0.9V_{br}$, and $V_{ces}$ = 0 V. There is a potential barrier $\Delta E_v$ between them, and it gradually increases from the surface to the inside of the device. The intrinsic InP cap layer between them is depleted at $d \leq 12$ μm. However, since this is a p-i-p structure, there is a back-to-back electric field in the depletion region, so the potential barrier is generated. (More details can be found in Supplementary Fig. 5.) We further selected the position at 100 nm below the surface (blue dotted line in Fig. 4c, $z$ = 6.92 μm) to analyze this potential barrier at the pixel bias voltage from 0 V to 55 V ($V_{br}$), the results are shown in Fig. 4d.

Potential barriers are marked with the dotted square boxes, and the $\Delta E_v$ at 55 V is clearly shown in the enlarged Fig. 4d. $\Delta E_v$ decreases with increasing pixel bias voltage, reaching 0.37 eV at $0.9V_{br}$. As a result of this low $\Delta E_v$, holes can flow from the CES to the pixel anode, resulting in the hole leakage current.

Reducing the value of $d$ results in a lower potential barrier and a larger hole leakage current, as shown in Fig. 4a, e. Figure 4e shows the $\Delta E_v$ when $d$ varies from 17 μm to 7 μm. A comparison of I−V characteristics in Fig. 4a and $\Delta E_v$ in Fig. 4e proves that the pixel and the CES can work independently when the $\Delta E_v$ is larger than -0.6 eV, the holes can be blocked by this barrier.

However, a small $d$ value is inevitable in a high pixel density array or a wide CES. To guarantee our CES can work properly in these arrays, we applied a larger $V_{ces}$ to increase $\Delta E_v$ to block the flow of holes. Figure 5a shows the I−V characteristics of the pixel when $d$ = 7 μm and the $V_{ces}$ varies from 0 V to 30 V. Figure 5b shows the $\Delta E_v$ at different $V_{ces}$, and Fig. 5c shows current density streamlines in the device cross-section at $0.9V_{br}$. A larger $\Delta E_v$ can be obtained by adopting a larger $V_{ces}$,

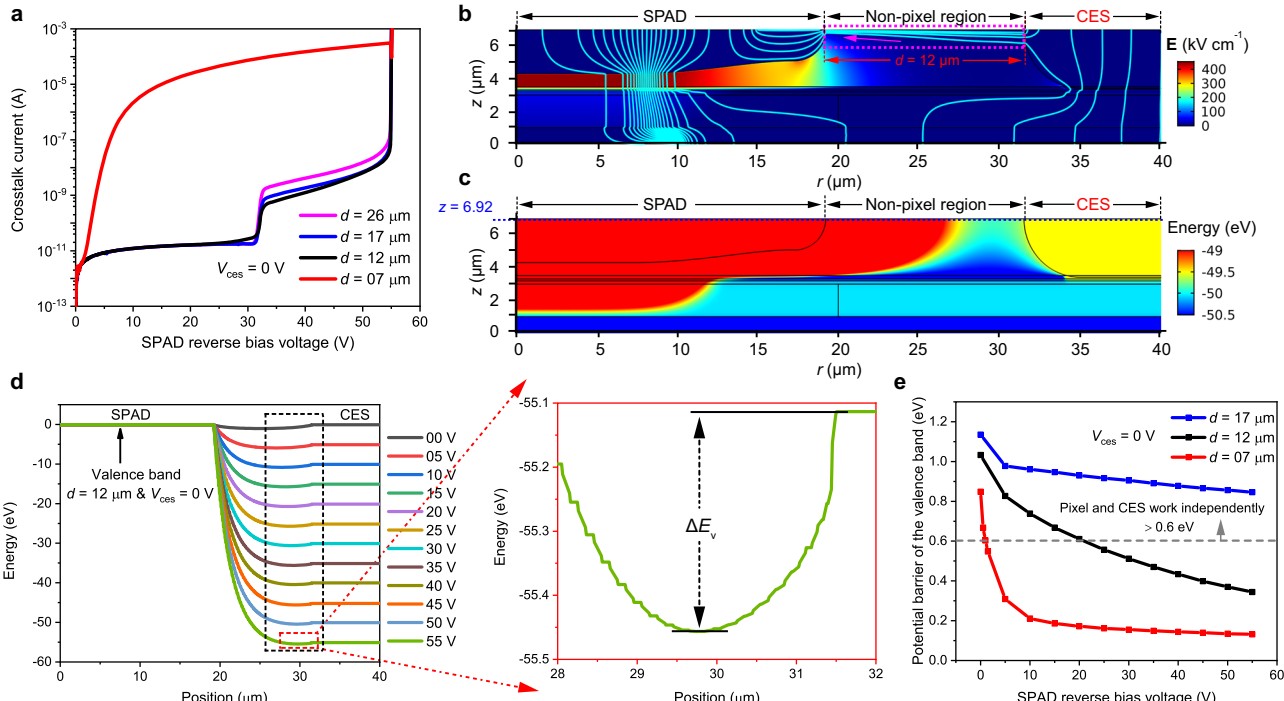

**Fig. 4 | Analysis of the leakage current between the single-photon avalanche diode (SPAD) and the carrier extraction structure (CES). a** Simulated crosstalk current-voltage ($I$–$V$) characteristics of the pixel in the array with the distances between the SPAD and the CES ($d$) varying from 26 μm to 7 μm, when the CES is biased at 0 V ($V_{ces} = 0$ V). **b** The flow chart of the crosstalk current density (cyan solid lines) when the pixel is biased at 90% breakdown voltage ($0.9V_{br}$, 49.5 V), at $d = 12$ μm and $V_{ces} = 0$ V. The figure also shows the electric field in the device cross-section surface (color legend on the right). Only the 0–40 μm region of the simulation structure is presented. **c**–**e** Source analysis of this hole leakage current. The distribution of the valence band **c** in the device cross-section at the pixel bias voltage of $0.9V_{br}$ and **d** at the position of 100 nm below the surface (blue dotted line in Fig. 4c, $z = 6.92$ μm) at different pixel bias voltages from 0 V to 55 V. **e** The potential barriers ($\Delta E_v$) at different pixel bias voltages while the distance $d$ varied from 17 μm to 7 μm.

as shown in Fig. 5b. Consequently, a $V_{ces}$ ~ 10 V is required in the array with $d = 7$ μm, as shown in Fig. 5a, c. Therefore, the measured results in Fig. 3e can be observed for the array with a 50 μm pixel pitch. A larger $V_{ces}$ can also obtain better crosstalk suppression, in Fig. 5a, the $N_{crosstalk}$ of $V_{ces} = 30$ V is decreased by 15.91% compared to $V_{ces} = 10$ V. The $V_{ces}$, however, cannot be too large. When it exceeds 30 V, a significant tunnel current generates in the InGaAs layer, which increases exponentially with the $V_{ces}$, reaching 10 μA at -39 V. Thus, to avoid CES breakdown, the maximum $V_{ces}$ is set to 39 V. Table 1 shows the bias conditions of the CES with different $d$, a larger $V_{ces}$ is required in the array with a smaller $d$. The simulated results are in agreement with the measured results. When $d > 12$ μm, the CES can realize high crosstalk suppression even it is biased at 0 V, but as the $d$ decreases to below 12 μm, a larger $V_{ces}$ is required. The smallest permissible $d$ value is ~3 μm.

In summary, numerical analysis and experimental measurements have proven that our CES can effectively suppress the optical-electrical crosstalk in planar InGaAs/InP SPAD arrays, achieving high crosstalk suppression without introducing any additional etching damage. And as $d$ decreases, the crosstalk suppression effect becomes better, as shown in Fig. 6. Therefore, for devices in Fig. 3a, we can further reduce the value of $d$ to fabricate arrays with lower crosstalk. Importantly, when $d$ is fixed, the crosstalk suppression effect of 2-μm-wide CES is the same as the 40-μm-wide CES. As a result, we can fabricate SPAD arrays with narrow CES and small pixel pitch to achieve high pixel density and low crosstalk. Considering the feasibility of the fabrication processes, the diffusion width $D$ of the CES is set to 2 μm. For the pixel with an active area diameter of 20 μm, when the CES works at zero bias voltage which is easy to operate, the pixel pitch can be reduced to ~74 μm; and when the CES operates at high reverse bias voltage, the pixel pitch can be further reduced to ~54 μm. In addition, we can

further optimize the pixel structures (reducing the active area diameter, optimizing the length of the shallow diffusion region, etc.) to obtain low-crosstalk planar InGaAs/InP SPAD arrays with higher pixel density. This CES has shown high crosstalk suppression in mainstream arrays with pixel pitches of 100/50 μm, and with the space to optimize to smaller pixel pitch. On the other hand, the optimized CES can also alleviate the influence on the dark count and the afterpulsing count caused by stray carriers returning to the active area, due to the photon absorption outside the working region.

## Discussion

In this work, by introducing an electric field-extracting CES between pixels, we demonstrate highly efficient crosstalk suppression in planar InGaAs/InP SPAD arrays. This CES is a p-doped channel that isolates pixels while guiding crosstalk carriers to the surface to suppress crosstalk via an applied electric field. For the fabricated arrays with 100 μm pixel pitch, a high crosstalk suppression of ~90% between the nearest neighbor pixels was observed because of the strong suppression of the optical-electrical crosstalk. Moreover, by narrowing the distance between the CES and pixels, better crosstalk suppression can be realized. Importantly, a narrow CES close to the pixel can considerably suppress crosstalk, this gives the space to further reduce the pixel pitch and thus raise the pixel density while achieving high crosstalk suppression. Currently, for the fabricated arrays with 50 μm pixel pitch, we have realized 96.22% high crosstalk suppression between the nearest neighbor pixels by adopting a 2-μm-wide CES. These results reveal that the optical-electrical crosstalk is the primary crosstalk mechanism for planar InGaAs/InP SPAD arrays, and the presented low-crosstalk SPAD arrays can be applied to high-resolution 3-D imaging and quantum techniques. Our work provides guidance for further understanding and suppression of crosstalk in InGaAs/InP

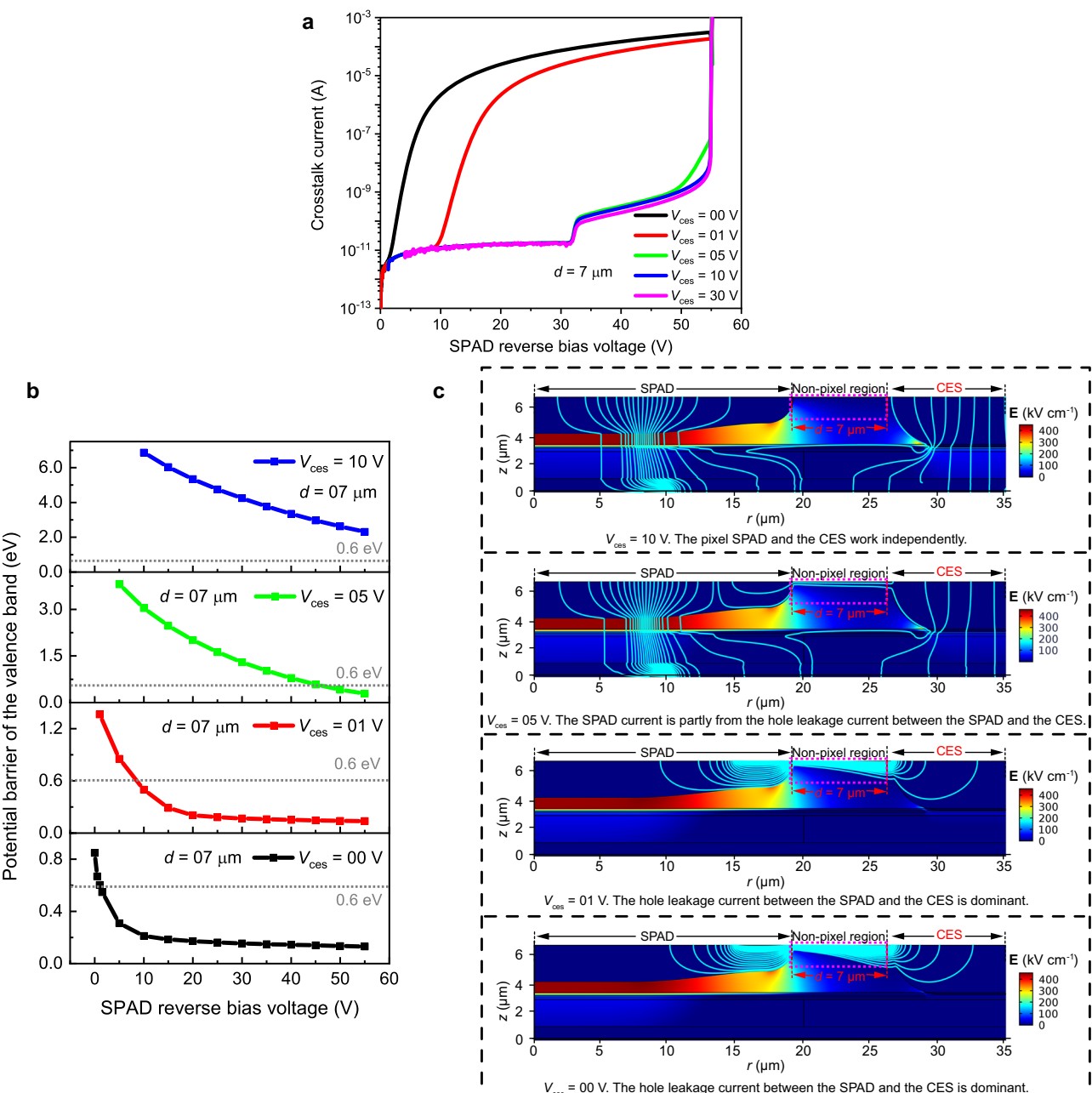

**Fig. 5 | Suppression and elimination of the hole leakage current between the single-photon avalanche diode (SPAD) and the carrier extraction structure (CES). a** Simulated crosstalk current-voltage (*I*–*V*) characteristics of the SPAD with the CES bias voltage ($V_{ces}$) varying from 0 V to 30 V, when the distance between the SPAD and the CES (*d*) is 7 μm. **b** The potential barriers ($\Delta E_v$) between the CES and the pixel at different pixel bias voltages ranging from 0 V to 55 V, and **c** the flow charts of the current density (cyan solid line) on the device cross-section at the pixel bias voltage of 90% breakdown voltage ($0.9V_{br}$, 49.5 V), when the $V_{ces}$ is varied from 0 V to 10 V. The color legends on the right show the electric field distribution of the device. Only the 0–35 μm region of the simulation structure is presented. A larger $V_{ces}$ is applied on the CES to increase the $\Delta E_v$, which can block hole flow from the CES to the pixel anode.

SPAD arrays. In addition, this CES can also be applied to Linear-mode APD-FPAs and other FPAs to isolate pixels and suppress crosstalk.

## Methods
### Device structure and simulation
The reported planar InGaAs/InP SPADs epitaxial structures were grown on two-inch n-type InP (100) substrates by MOCVD. The lattice-matched n-doped InP buffer layer, intrinsic $In_{0.53}Ga_{0.47}As$ absorption layer, intrinsic InGaAsP grading layer, n-doped InP charge layer, and intrinsic InP cap layer were sequentially grown on InP substrates. 2-μm-thick $In_{0.53}Ga_{0.47}As$ layer to obtain high external quantum efficiency (EQE). 3 layers of InGaAsP with the thickness of

120 nm (3 × 40 nm) alleviate the potential barrier of the valence band between the $In_{0.53}Ga_{0.47}As$ layer and InP charge layer, reducing the accumulation of photogenerated holes at the heterojunction interface. The surface charge density of the 200-nm-thick InP charge layer was adjusted to ~2.8 × $10^{12}$ cm$^{-2}$, and the high electric field mainly falls in the InP multiplication layer to trigger avalanche gain; simultaneously, the electric field in the $In_{0.53}Ga_{0.47}As$ absorption layer is controlled (<150 kV cm$^{-1}$) to make photogenerated holes drift at the saturation velocity, but no obvious tunneling. The P-N junction of the pixel was formed by double Zn diffusion in a MOCVD reactor in the 3.5-μm-thick InP cap layer, which also defines the active area diameter and the multiplication layer thickness. In the array with a

**Table 1 | The selection of the reverse bias voltage ($V_{ces}$) applied on the carrier extraction structure (CES) to ensure that the pixel and the CES work independently at the different distances ($d$)**

| Symbol | $d$ (µm) | $V_{ces}$ (V) |
|---|---|---|
| Simulated | >12 | 0–39 |
| | 12 | 1–39 |
| | 7 | 10–39 |
| | 3 | 35–39 |
| Experimental | 26 | 0–39[a] |
| | 6 | 10–39[b] |

39 V is the maximum reverse bias voltage applied on the CES. Measured results in arrays with a 100 µm pixel pitch[a] and a 50 µm pixel pitch[b].

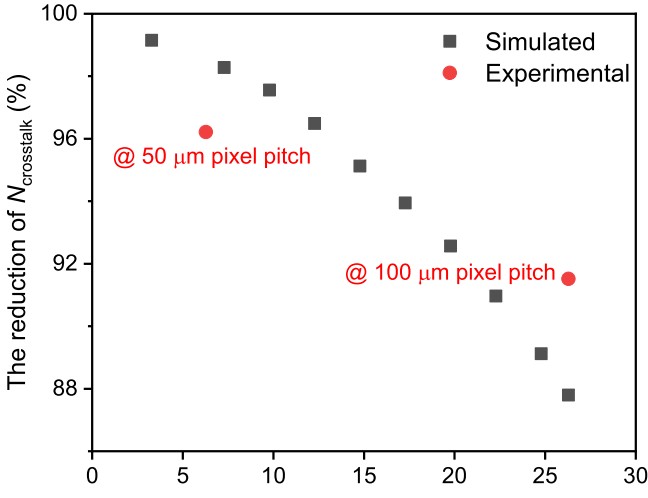

**Fig. 6 | The crosstalk suppression effect in arrays with different distances between the pixel and the carrier extraction structure (CES).** The figure shows the results of the reduction in the number of carriers collected by the *detector* ($N_{crosstalk}$) in the simulation (black squares) and experimental (red circles). A CES that is closer to the pixel can achieve better crosstalk suppression.

100 µm pixel pitch, the active area diameter is 20 µm, and 10 µm for the array with a 50 µm pixel pitch. The difference between the radius of the shallow diffusion junction and the center deep diffusion junction is 8 µm. The CES was fabricated by an additional MOCVD Zn deep diffusion. In the arrays without CES, the pixel structures adopted a combination of double diffusion and floating guard ring (FGR); but, the FGR was removed in arrays with CES. Since the FGR primarily reduces the surface electric field and minimizes the surface leakage current, while the surface current does not contribute to dark counts; and the uniform distribution of the avalanche electric field in the active area can be realized by optimizing the double Zn diffusion and active area diameter[29,31]. At the same time, our measured results also show that the FGR has almost no suppression effect on the crosstalk (Supplementary Fig. 8). Therefore, we have removed the FGR in our newly designed pixel, which is beneficial for further increasing the pixel density.

In the simulation analysis, we developed a two-dimensional (2D) rotationally symmetric model in the finite element analysis software (COMSOL Multiphysics) based on the above structural parameters. In the simulation model, the contour of the p-doped InP diffusion region (both pixel and CES region) was extracted according to the experimental diffusion contour from scanning electron microscope (SEM) images (the typical diffusion profiles can be found in Supplementary

Fig. 3). And for InP impact ionization, we employed the Zappa impact ionization model, which is related to the electric field intensity and temperature, with the relevant parameters reported in[34].

## Device fabrication

A 200-nm-thick silicon nitride ($SiN_x$) was grown on the InP cap layer by Plasma Enhanced Chemical Vapor Deposition (PECVD) as Zn diffusion mask. Then the diffusion windows were opened by photolithography and ICP etching processes. The Zn diffusion temperature was set to 530 °C and the diffusion time was selected according to the diffusion depth. The order of three Zn diffusion processes was the deep diffusion of the CES, the shallow diffusion of the pixel and the FGR, and the central deep diffusion of the pixel. The deep diffusion depth of the CES was 3.7 µm (total thickness of the InP cap layer and InP charge layer). The total diffusion depth of the pixel was about 2.5 µm, forming a 1-µm-thick InP multiplication layer. The double diffusion step of the pixel was about 400 nm. After each Zn diffusion, we used $H_2SO_4$: $H_2O_2$: $H_2O$ (3:1:1) to clean the epitaxial wafer and HF: $H_2O$ (1:1) to remove the $SiN_x$ diffusion mask. Then, a new 200-nm-thick $SiN_x$ was deposited on the top surface as the passivation layer. The top p-contact metallization of pixel and CES was made of 30/40/250-nm-thick Ti/Pt/Au hybrid thin films. Then, the InP substrate was thinned and polished to ~250 µm. A 200-nm-thick and $n = 2$ $SiN_x$ antireflection film was deposited on the bottom. Finally, 150/250-nm-thick Au-Ge-Ni/Au hybrid thin films were deposited to form a common n-contact metallization on the bottom.

## DC crosstalk measurements

In this paper, from the perspective of photon detection efficiency (PDE) and dark count rate (DCR), we proposed that the crosstalk characteristics between SPADs can be analyzed from the variation of crosstalk carriers collected by the device at unity gain. The method based on pixel DC $I$–$V$ characteristics to analyze crosstalk was proposed, called DC crosstalk measurements.

The DCR of a SPAD is determined by the product of dark carrier generation rate and avalanche triggering probability ($P_{avl}$)[27]. The PDE is determined by the product of external quantum efficiency (EQE) and $P_{avl}$[35,36]

$$PDE = EQE \times P_{avl} \qquad (1)$$

The $P_{avl}$ is mainly determined by the $V_{ex}$. The EQE can be estimated by measured $I$–$V$ results, at unity gain the EQE can be calculated by

$$EQE = \frac{I_p}{P_{opt}} \times \frac{h\upsilon}{e} \qquad (2)$$

Where $I_p$ is the measured photocurrent at unity gain, $P_{opt}$ is the incident optical power, $h\upsilon$ is the power of the incident photon, and $e$ is the elementary charge.

The essence of generating a crosstalk count is the same as the photon count or dark count, as both the InP multiplication layer captures one or multiple holes to trigger self-sustaining avalanche gain. Therefore, the crosstalk count rate (CCR) of the *detector* can be determined by the product of crosstalk carrier generation rate ($N_{crosstalk}$) and $P_{avl}$. The CCR can be written as

$$CCR = N_{crosstalk} \times P_{avl} \qquad (3)$$

Crosstalk can be further characterized by the number of crosstalk carriers collected by the *detector*, which is the fundamental source of the crosstalk count. Analysis of this essential parameter avoids the influence of numerous parameters in Geiger mode, such as excess bias voltage, gating frequency, quenching resistance, etc., and provides stable results and correct judgements. From the Eq. (2), $N_{crosstalk}$ can be

estimated from the $I-V$ measurements. At unity gain, the $N_{crosstalk}$ is determined by

$$N_{crosstalk} = \frac{I_v - I_d}{e} \qquad (4)$$

Where $I_v$ and $I_d$ represent the measured crosstalk and dark currents at unity gain. Under the real pulse count, the variation of $I_v$ at unity gain is negligible. Therefore, we further proposed the DC crosstalk measurements. In DC crosstalk measurements, the $I-V$ characteristics of the *detector* are obtained in the dark environment with another adjacent pixel as the crosstalk source (*emitter*). This *emitter* has been in a self-sustaining avalanche state, biased above the breakdown at a constant current (>10 μA). A value proportional to the crosstalk count rate is obtained by Eq. (4) when the *emitter* is ON and OFF. The $I_d$ of the *detector* is obtained when the *emitter* is OFF, and the $I_v$ is obtained when the *emitter* is kept ON. Evaluating the crosstalk of this constant crosstalk source has the same effect as the real pulse crosstalk[30,37], but this method is more high-efficiency. And we have also developed relevant measurements and demonstrated that the measured crosstalk suppression of the CES is basically the same between the DC crosstalk measurements and the pulse counting mode.

**Crosstalk measurement setup details**

We mainly adopted the DC crosstalk measurements to characterize the crosstalk characteristics between the nearest neighbor pixels in the fabricated 64 × 64 arrays (as shown in Fig. 3a). The entire test processes were performed on a three-dimensional movable probe station. First, the $I-V$ characteristics of devices were measured in dark conditions and when illuminated with a continuous wave laser (λ = 1550 nm, KEYSIGHT N7714A), respectively. According to the measured results, two nearest neighbor pixels that work well were selected as devices under test. One pixel was selected as the *emitter* (crosstalk source) to emit crosstalk photons. Another pixel as the *detector* (victim pixel) collects crosstalk carriers to achieve crosstalk characterization. In the array without CES, the *emitter* was operated at a constant current (from 100 μA to 1000 μA) by a DC voltage source (KEITHLEY 2450 Source-Meter), then the crosstalk current curves of the *detector* were scanned by a semiconductor characterization system (KEITHLEY 4200 – SCS). The *detector* current is limited to 10 μA during the scanning process. In the array with CES, firstly, a constant reverse bias voltage $V_{ces}$ was applied on the CES by a DC voltage source (KEITHLEY 2400 Source-Meter), and the *emitter* was operated at a constant current by another DC voltage source (KEITHLEY 2450 SourceMeter), then the crosstalk current curves of the *detector* were scanned by KEITHLEY 4200 – SCS. Finally, the crosstalk characteristics were obtained by Eq. (4). And then the crosstalk suppression effect of the CES can be obtained by comparing the variation of $N_{crosstalk}$ in arrays without and with CES. (More crosstalk measurement setup details can be found in Supplementary Fig. 7).

Crosstalk suppression of the CES in pulse counting mode was obtained by operating the *detector* in Geiger mode, this method was also called pseudo-crosstalk measurements[37]. The *detector* was operated with gate bias at 12.5 ns pulse width and 100 kHz repetition rate, which eliminates the influence of afterpulse (hold-off time -10 μs). The DC bias was kept 1 V below its breakdown voltage during OFF-time, while during ON-time an excess bias voltage ($V_{ex\_D}$) of 0.5 V was applied. The influence of spike noise was eliminated by a balanced APD. The counter is SR400 photon counter, and the counting period was set to 1 s. The average of dark counts and total counts (dark counts + crosstalk counts) was obtained by recording multiple data sets separately. When the *emitter* was turned OFF, the dark counts of the *detector* were recorded; and when the *emitter* was kept at a constant current, the total counts were recorded. Then, the crosstalk counts were obtained by subtracting the dark counts from the total counts.

The test was repeated several times during a week to observe the fluctuation of the CES crosstalk suppression.

## Data availability

All the data supporting the findings of this study are available within the article and its Supplementary Information file. Additional data are available from the corresponding author upon request.

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

## Acknowledgements

X.Y. acknowledges the National Natural Science Foundation of China (61774152). We thank X.M. for part of the material processing. We thank the members of the Engineering Research Center for Semiconductor Integrated Technology, Institute of Semiconductors, CAS for assistance in device fabrication. We sincerely thank all authors for their assistance in device simulation and fabrication, test analysis, and manuscript writing.

## Author contributions

X.Y. conceived this idea. Y.T. and X.Y. designed and fabricated the devices, implemented the experiments, and developed analysis methods. R.W., T.H., and Y.L. contributed to the numerical simulation of device structures and device fabrication. R.W. and M.Z. contributed to data characterization. Y.T. and X.Y. wrote and revised the manuscript. All authors made substantial contributions to this research.

## Competing interests

The authors declare no competing interests.
