## [Peer Review File · Nature Communications]

REVIEWER COMMENTS

Reviewer #1 (Remarks to the Author):

General comments:

The authors present a study on a structure called “CES - carrier extraction structure” to be employed in InGaAs/InP SPAD arrays for reducing crosstalk between neighboring pixels. Such CES should avoid the need for etched trenches for crosstalk suppression and it is therefore of interest to the community working on InGaAs/InP SPADs.

However, there is no direct comparison between the etched trenches and the carrier extraction structure on the overall effectiveness in suppressing crosstalk. It is claimed that the proposed CES approach is 1.5 better than etched trenches, but such comparison is made with data reported in literature from SPADs with different layer thicknesses, which lead to different electric field distribution. A direct comparison between CES and etched trenches is therefore required for assessing their relative (and, maybe, also combined?) efficiency in suppressing crosstalk. If this is not possible, I think that no direct comparison can be made and the “1.5-fold improvement” statement has to be removed from this manuscript.

A weak point of this work is that crosstalk suppression via CES is evaluated by operating the detector as an APD, in linear mode (i.e., below breakdown), and not as a SPAD, in Geiger mode (i.e., beyond breakdown). Given the non-uniform and non-linear relationships between the electric field and the avalanche triggering probability, the results from the photogenerated currents might not be representative of the behavior in photon counting mode. Since the device under test is a SPAD, why no measurement is presented while it is operated as a SPAD? The CES performance should be evaluated by using the avalanche rates.

Especially with a very thick (2 μm) InGaAs layer, as the one employed for the detectors under test, floating guard ring (FGR) are typically employed: why no FGR is included? What is its impact on the crosstalk? (also with reference to line 379-380)

English should be revised and improved throughout the manuscript. Also, a few statements and comments are repeated more times throughout the manuscript.

Detailed comments:

Abstract and Introduction: It should be clearly stated that the CES structure here presented is not useful for all SPAD arrays (including the widespread silicon ones), but just for InGaAs/InP SPAD arrays, or for similar ones where absorption and multiplication materials are different.

Fig. 3: Please, report in the figure / caption the operating temperature.

From line 326 to 331: please, compare these numbers with those from fabricated InGaAs/InP SPAD arrays with trenches reported in literature.

Line 335: Instead of “fluorescence absorption”, please refer to “photon absorption”.

Supplementary information:

Fig. S1: Why did you consider only the vertical electric field, neglecting its horizontal component?

Reviewer #2 (Remarks to the Author):

The manuscript entitled "High crosstalk suppression in InGaAs/InP single-photon avalanche diode arrays by carrier extraction structure" by Yongsheng Tang et al., submitted for publication in nature communications has been reviewed.

In this manuscript, the authors report on a new crosstalk suppression method. They demonstrated that a carrier extraction structure (CES) between pixels can reduce crosstalk in both simulation and experimental measurements. Since the crosstalk is one of the most important issues for single-photon avalanche diode arrays for any applications, the realized suppression ratio of more than 90% or 96% (depending on the pixel pitch) is an important achievement in this research field. Remarkably, the large suppression effect reveals that optical-electrical crosstalk is the dominant source rather than optical crosstalk. As well as showing that the CES can suppress crosstalk, they have done a comprehensive study on the suppression mechanism in detail.

I noticed several points that must be first considered to improve the manuscript.

1. In this manuscript, the crosstalk has been studied by a method which is called DC crosstalk measurement. Although they have justified that this method has the same effect as the real pulse crosstalk by several mathematics in the section "DC crosstalk measurements", it is better to show some experimental evidence of the equableness since the study highly relies on this method. It would be useful if appropriate references are added if there would be. Or, it would be appreciated that supplemental measurements that compare the real pulse crosstalk and DC crosstalk. That supports the statement in lines 441-442.

2. It is not clear whether the simulation conducted in the manuscript is considering the conventional optical crosstalk or not. Please clarify that. Otherwise, readers cannot be sure whether the suppression effect shown in the simulations is total crosstalk suppression or optical-electrical crosstalk suppression only.

3. The authors claim that the dominant crosstalk source in InGaAs arrays has turned out to be optical-electrical crosstalk, rather than optical crosstalk which was considered as the main crosstalk mechanism. Is it due to the absorption layer or other mechanisms?

Also, could you mention the case of Si arrays if possible? I am curious if it is the same or different.

4. It is good to show the absolute crosstalk probability in the array (the probability of a secondary crosstalk pulse generation with a first pulse generation in the array) as well as the suppression ratio, like reference 22 and others.

5. Figure 4 is a bit busy. Could you consider separating these subfigures into two figures and expanding them?

6. Minor comment

- Subscript characters (e.g. CES in V_{CES}) are roman font.

Reviewer #1 (Remarks to the Author):

Response: We sincerely appreciate your review and valuable comments on our manuscript. The manuscript has been revised based on your suggestions. All revisions have been highlighted in yellow in the revised manuscript. Below is a point-by-point response to your comments.

General comments:

The authors present a study on a structure called “CES - carrier extraction structure” to be employed in InGaAs/InP SPAD arrays for reducing crosstalk between neighboring pixels. Such CES should avoid the need for etched trenches for crosstalk suppression and it is therefore of interest to the community working on InGaAs/InP SPADs.

However, there is no direct comparison between the etched trenches and the carrier extraction structure on the overall effectiveness in suppressing crosstalk. It is claimed that the proposed CES approach is 1.5 better than etched trenches, but such comparison is made with data reported in literature from SPADs with different layer thicknesses, which lead to different electric field distribution. A direct comparison between CES and etched trenches is therefore required for assessing their relative (and, maybe, also combined?) efficiency in suppressing crosstalk. If this is not possible, I think that no direct comparison can be made and the “1.5-fold improvement” statement has to be removed from this manuscript.

Response: The reviewer is correct. As the reviewer described, the epi-layer thickness, electric field distribution, and test conditions all have a direct effect on measurements, so that a reliable comparison of the suppression effect between CESs and etched trenches is difficult for devices fabricated in two groups. Therefore, we have removed the “1.5-fold improvement” statement in the revised manuscript text on Page 2, lines 29-31, and Page 14, lines 283-285. We have also done relevant experiments to investigate the crosstalk suppression by etched trenches, but since the device structures are different from those presented in the manuscript, we did not report them in this work.

A weak point of this work is that crosstalk suppression via CES is evaluated by operating the detector as an APD, in linear mode (i.e., below breakdown), and not as a SPAD, in Geiger mode (i.e., beyond breakdown). Given the non-uniform and non-linear relationships between the electric field and the avalanche triggering probability, the results from the photogenerated currents might not be representative of the behavior in photon counting mode. Since the device under test is a SPAD, why no measurement is presented while it is operated as a SPAD? The CES performance should be evaluated by using the avalanche rates.

Response: The reviewer has raised a concern as to why we did not evaluate the CES performance in Geiger mode. Indeed, the avalanche count is a key parameter for evaluating the performance of SPADs. But we also believed that the number of free carriers, which is the intrinsic source of the avalanche count, can also be an important parameter for evaluating the performance of SPADs or CESs. Theoretically, each carrier that makes up the current passing through the multiplication region

would participate in avalanche multiplication with an avalanche probability. Analysis of this current can avoid the influence of numerous parameters in Geiger mode, such as excess bias voltage, gating frequency, quenching resistance, etc., and can provide stable results and correct judgements. Therefore, we quantify the CES performance from the number of carriers injected into the InP multiplication layer.

As the reviewer was concerned, considering the non-uniform and non-linear relationship between the electric field and the avalanche triggering probability, the results of the DC crosstalk measurements may differ from the avalanche counting mode. Therefore, to evaluate the CES performance in pulse counting mode, we fabricated new arrays on the same wafer. Their measured results are shown in Fig. 1. In DC measurements, the CES shows $\sim 90.47\%$ crosstalk suppression at $V_{ces} = 0$ V, and $\sim 91.52\%$ at $V_{ces} = 10$ V, when the *emitter* working current is 200 μ A, as shown in Fig. 1a. When the *detector* is operated in Geiger mode, the CES shows an average crosstalk suppression of $\sim 88.60\%$ at $V_{ces} = 0$ V, and 89.82% at $V_{ces} = 10$ V, as shown in Fig. 1b. The CES performance in pulse counting mode is basically the same as the DC crosstalk measurements with a little difference.

We have added the results in pulse counting mode to Fig. 3 (Fig. 3d); the legend has been added on Pages 25-26, lines 529-531. And a discussion and the test setup of the CES performance in pulse counting mode has also been provided on Page 10, lines 202-206 (Results); Pages 19-20, lines 402-413 (Methods).

In addition, to compare the CES performance at the same *emitter* working current (200 μ A), we have also replaced the crosstalk suppression effect of the DC crosstalk measurements in the revised manuscript on Page 2, line 30; Page 9, lines 185-187, 194-195; and Fig. 6.

Fig. 1 The measured crosstalk suppression of the CES between the nearest pixels at 300 K. Measured results **a** when the *detector* was operated in Linear mode; and **b** when the *detector* was operated in Geiger mode, the *detector* was operated in gated mode at a repetition frequency of 100 kHz (hold-off time is ~ 10 μ s and the effect of afterpulse can be neglected) and 0.5 V overbias. The error bar represents the fluctuation in counts between repeated tests.

Other experimental results also demonstrated that the external quantum efficiency (EQE) enhancement measured in Linear mode was basically the same as the photon detection efficiency (PDE) improvement in Geiger mode, as presented by Zhang, B. et al.¹. Relevant figures are reproduced, as shown in Fig. 2. Compared with SPAD#2, which has the same active area diameter

but without the quantum efficiency enhancement design, the EQE of SPAD#1 was increased by 58.4% by integrating the DBR-metal reflector and backside micro-lens, as shown in Fig. 2a. And for the same normalized dark count rate (DCR), the PDE of SPAD#1 is basically 1.58 times that of SPAD#2, as shown in Fig. 2b. We have also cited Zhang, B. et al. on Page 18, line 379 in the revised manuscript.

Fig. 2 Reproduced from Zhang, B. et al. **a** Measured I - V characteristics at room temperature; **b** normalized DCR versus PDE at 233 K of different SPADs. The inset is the rescaled curves of SPAD#1 and SPAD#2, where the PDE of SPAD#1 is divided by 1.58 (the ratio of optical absorption of SPAD#1 and SPAD#2).

Especially with a very thick (2 μm) InGaAs layer, as the one employed for the detectors under test, floating guard ring (FGR) are typically employed: why no FGR is included? What is its impact on the crosstalk? (also with reference to line 379-380)

Response: The reviewer correctly pointed out that we lacked an explanation for the no floating guard ring (FGR) design and the impact of FGR on the crosstalk. Double Zn diffusion and FGR are the structures typically employed in planar InGaAs/InP APDs and SPADs. The FGR primarily reduces the surface electric field and minimizes the surface leakage current, while the surface current does not contribute to the avalanche count. And our work² and Signorelli, F. *et al*³ have demonstrated that the uniform distribution of the avalanche electric field in the active area can be realized by optimizing the double Zn diffusion and the active area diameter. At the same time, our measurement results also show that the FGR has almost no suppression on the crosstalk, as shown in Fig. 3. Therefore, we have removed the FGR in our newly designed pixel, which is also beneficial for further increasing the pixel density.

An explanation has been added in Methods section on Page 15, lines 315-321 in the revised manuscript. And we have also added the measurement results and discussion to Supplementary information (Supplementary Fig. 8, Page 10, lines 160-165).

Fig. 3 Measurement results of the impact on the crosstalk by floating guard ring (FGR). The solid lines represent the measured I - V characteristics and the dashed lines represent the voltage dependence of the gain (M - V).

English should be revised and improved throughout the manuscript. Also, a few statements and comments are repeated more times throughout the manuscript.

Response: Thank you very much for pointing this out. We have carefully proofed the manuscript, and made several grammatical and formatting revisions. The English has been improved in the revised manuscript and Supplementary information.

Detailed comments:

Abstract and Introduction: It should be clearly stated that the CES structure here presented is not useful for all SPAD arrays (including the widespread silicon ones), but just for InGaAs/InP SPAD arrays, or for similar ones where absorption and multiplication materials are different.

Response: We believe that the CES is also useful in other SPAD arrays, it may be similar to the P-type guard in Si arrays^{4,5}, but the CES does have better performance in the device with different absorption and multiplication materials. We have added "It should be noted that this technique can also be used in other SPAD arrays, but works better in arrays with different absorption and multiplication materials." on Page 4, lines 87-88 (Introduction) in the revised manuscript.

Fig. 3: Please, report in the figure / caption the operating temperature.

Response: The operating temperature in Fig. 3 is 300 K, we have added it to Fig. 3; Page 8, line 173 (Results); Page 25, line 523 (legend for Fig. 3) in the revised manuscript. And we have also added the temperature (300 K) set for the numerical simulation to Fig. 2; Page 25, line 515 (legend for Fig. 2).

From line 326 to 331: please, compare these numbers with those from fabricated InGaAs/InP SPAD arrays with trenches reported in literature.

Response: The reviewer correctly pointed out that we lack comparisons with reported pixel densities using etched trenches. However, since the reported devices have different active area diameters, shallow diffusion lengths, etc., they are not suitable for a direct comparison. Therefore, in the manuscript, we mainly show what can be realized for our device. We have modified the description in this paragraph (Page 13, lines 269-272). And we have also added "This CES has shown high crosstalk suppression in mainstream arrays with pixel pitches of 100/50 μm , and with the space to optimize to smaller pixel pitch." on Page 13, lines 274-276.

Line 335: Instead of “fluorescence absorption”, please refer to “photon absorption”.

Response: We have replaced “fluorescence absorption” with “photon absorption” on Page 13, line 277-278 in the revised manuscript.

Supplementary information:

Fig. S1: Why did you consider only the vertical electric field, neglecting its horizontal component?

Response: We appreciate the reviewer pointing out that we neglected the horizontal electric field component. The simulation results show that the horizontal electric field mainly distributes at the edges of the active area and the shallow diffusion region; and the non-pixel region is a neutral region (the horizontal electric field is 0 kV cm^{-1}), as shown in Fig. 4. Therefore, the main contribution to direct electrical crosstalk is the diffusion of minority hot carriers (holes) in the neutral region. However, for double diffusion InGaAs/InP SPADs, there is a shallow diffusion region with the width of 8-10 μm outside the active area, where the electric field can recapture avalanche hot carriers that have diffused out, so that minority hot carriers cannot diffuse to adjacent pixels, avoiding the influence of the direct electrical crosstalk.

Fig. 4 The distribution of the horizontal electric field component in the device at 1 V excess bias voltage.

We have added horizontal electric field to Supplementary Fig. 1b, the legend and a discussion have also been added on Page 3, lines 47-48, and lines 52-56 in the revised Supplementary information.

References

1. Zhang, B. et al. High Performance InGaAs/InP Single-Photon Avalanche Diode Using DBR-Metal Reflector and Backside Micro-Lens. *J. Lightwave Technol.* **40**, 3832-3838(2022).
2. He, T., Yang, X., Tang, Y., Wang, R. and Liu, Y. Quantitative analysis of edge breakdown effect of Geiger mode avalanche photo-diodes utilizing optical probe scanning method. *Semicond. Sci. Technol.* **37**, 055006 (2022).
3. Signorelli, F. et al. Low-Noise InGaAs/InP Single-Photon Avalanche Diodes for Fiber-Based and Free-Space Applications. *IEEE J. Sel. Topics Quantum Electron.* **28**, 1-10 (2022).
4. Vilà, A., Vilella, E., Alonso O. and Dieguez, A. Crosstalk-Free Single Photon Avalanche Photodiodes Located in a Shared Well. *IEEE Electron Device Letters*, 35, 99-101 (2014).
5. Buttafava, M. et al. SPAD-based asynchronous-readout array detectors for image-scanning microscopy. *Optica* **7**, 755-765 (2020).

Reviewer #2 (Remarks to the Author):

The manuscript entitled "High crosstalk suppression in InGaAs/InP single-photon avalanche diode arrays by carrier extraction structure" by Yongsheng Tang et al., submitted for publication in nature communications has been reviewed.

In this manuscript, the authors report on a new crosstalk suppression method. They demonstrated that a carrier extraction structure (CES) between pixels can reduce crosstalk in both simulation and experimental measurements. Since the crosstalk is one of the most important issues for single-photon avalanche diode arrays for any applications, the realized suppression ratio of more than 90% or 96% (depending on the pixel pitch) is an important achievement in this research field. Remarkably, the large suppression effect reveals that optical-electrical crosstalk is the dominant source rather than optical crosstalk. As well as showing that the CES can suppress crosstalk, they have done a comprehensive study on the suppression mechanism in detail.

I noticed several points that must be first considered to improve the manuscript.

Response: Thank you very much for your valuable remarks on our manuscript. The manuscript has been revised based on your suggestions. All revisions have been highlighted in yellow in the revised manuscript. Below is a point-by-point response to your remarks.

1. In this manuscript, the crosstalk has been studied by a method which is called DC crosstalk measurement. Although they have justified that this method has the same effect as the real pulse crosstalk by several mathematics in the section "DC crosstalk measurements", it is better to show some experimental evidence of the equableness since the study highly relies on this method. It would be useful if appropriate references are added if there would be. Or, it would be appreciated that supplemental measurements that compare the real pulse crosstalk and DC crosstalk. That supports the statement in lines 441-442.

Response: The reviewer has correctly pointed out that we should have shown that the DC crosstalk measurements have the same effect on the real pulse crosstalk. In the revised manuscript, we have provided new experimental data to support this point.

To compare the CES performance in DC crosstalk measurements and pulse counting mode, we fabricated new arrays on the same wafer. The measurement results are shown in Fig. 1. In DC measurements, the CES shows ~90.47% crosstalk suppression at $V_{ces} = 0$ V, and ~91.52% at $V_{ces} = 10$ V, when the *emitter* working current is 200 μ A, as shown in Fig. 1a. When the *detector* is operated in Geiger mode, the CES shows an average crosstalk suppression of ~88.60% at $V_{ces} = 0$ V, 89.82% at $V_{ces} = 10$ V, as shown in Fig. 1b. The CES performance in pulse counting mode is basically the same as DC measurements with a little difference.

We have added the measured results in pulse counting mode to Fig. 3 (Fig. 3d); the legend has been added on Pages 25-26, lines 529-531. And a discussion and the test setup about the CES performance in pulse counting mode has also been provided on Page 10, lines 202-206 (Results);

Pages 19-20, lines 402-413 (Methods). In addition, to compare the CES performance at the same emitter working current (200 μA), we have also replaced the crosstalk suppression effect of the DC crosstalk measurements in the revised manuscript on Page 2, line 30; Page 9, lines 185-187, 194-195; and Fig. 6.

Fig. 1 The measured crosstalk suppression of the CES between the nearest pixels at 300 K. Measured results **a** when the *detector* was operated in Linear mode; and **b** when the *detector* was operated in Geiger mode, the *detector* was operated in gated mode at a repetition frequency of 100 kHz (hold-off time is $\sim 10 \mu\text{s}$ and the effect of afterpulse can be neglected) and 0.5 V overbias. The error bar represents the fluctuation in counts between repeated tests.

On the other hand, in reference 30, Zhang, B. et al. reported results show that the enhancement of the external quantum efficiency (EQE) measured in Linear mode was basically equivalent to the improvement of the photon detection efficiency (PDE) in Geiger mode. The relevant figures from reference 30 are reproduced below, as shown in Fig. 2. Compared with SPAD#2, which has the same active area diameter but without the quantum efficiency enhancement design, the EQE of SPAD#1 was increased by 58.4% by integrating the DBR-metal reflector and backside micro-lens, as shown in Fig. 2a. And for the same normalized dark count rate (DCR), the PDE of SPAD#1 is basically 1.58 times that of SPAD#2, as shown in Fig. 2b. We have also added reference 30 on Page 18, line 379 in the revised manuscript.

Fig. 2 Reproduced from Zhang, B. et al. (reference 30). **a** Measured I - V characteristics at room temperature; **b** normalized DCR versus PDE at 233 K of different SPADs. The inset is the rescaled curves of SPAD#1 and SPAD#2, where the PDE of SPAD#1 is divided by 1.58 (the ratio of optical absorption of SPAD#1 and SPAD#2).

2. It is not clear whether the simulation conducted in the manuscript is considering the conventional optical crosstalk or not. Please clarify that. Otherwise, readers cannot be sure whether the suppression effect shown in the simulations is total crosstalk suppression or optical-electrical crosstalk suppression only.

Response: The simulations in the manuscript considered the effects of both conventional optical crosstalk and optical-electrical crosstalk, so the suppression effect shown in the simulations is the total crosstalk suppression effect. Now, we have added “Here both the conventional optical crosstalk and the optical-electrical crosstalk were considered, so the simulation results show the total crosstalk and crosstalk suppression.” on Page 6, lines 117-119 in the revised manuscript.

3. The authors claim that the dominant crosstalk source in InGaAs arrays has turned out to be optical-electrical crosstalk, rather than optical crosstalk which was considered as the main crosstalk mechanism. Is it due to the absorption layer or other mechanisms?

Response: It is mainly caused by the absorption of crosstalk photons in the InGaAs layer between planar diffusion SPADs. In InGaAs/InP SPADs, the absorption and multiplication materials are different, in that the InGaAs absorption layer has a narrower band gap and a larger cut-off wavelength. Our measured results in Fig. 3b,f of the manuscript show that the crosstalk current and the dark current are basically the same before the SPAD punch-through, while the crosstalk current increases dramatically near the punch-through point, which proves that the absorption of long-wavelength crosstalk photons in InGaAs layer is the main contribution. However, by calculation, the direct optical absorption is minimal in such a pixel pitch. And if the optical crosstalk is considered to be the main contribution, the effect of our CES on crosstalk suppression can be negligible; but both simulations and experimental results show that the crosstalk decreases rapidly after the introduction of the CES, which supports our conclusion that the optical-electrical crosstalk is the main source of crosstalk in planar InGaAs/InP arrays. We have explained the cause of the optical-electrical crosstalk on Page 4, lines 68-71 (Introduction).

Also, could you mention the case of Si arrays if possible? I am curious if it is the same or different.

Response: The results are different for Si SPAD arrays. For InGaAs/InP and Si arrays, the main difference is that InGaAs/InP SPADs have different absorption and multiplication materials and there is a larger cut-off wavelength in the narrow band absorption layer; while materials in Si SPADs are the same. For Si arrays, most of the sensitive crosstalk photons are absorbed in the N-well (or P-well)^{1,2,3}, which contributes little to the crosstalk; only a very small number of crosstalk photons can cause optical crosstalk. For InGaAs/InP arrays, the absorption of short-wavelength crosstalk photons in the neutral InP cap layer is similar to that of Si arrays, which contributes little to the crosstalk; while the absorption of long-wavelength crosstalk photons in the InGaAs layer would be the main crosstalk mechanism. This is also the main reason why the crosstalk probability of InGaAs/InP arrays is much larger than Si arrays^{1,3,4}; at the pixels distance of 120 μm and 5 V overbias, the crosstalk probability of the InGaAs/InP array is $\sim 90\%$ ⁴, however, it is only $\sim 0.5\%$ for the Si array¹.

We have added “This is different from Si arrays, the main reason is that the InGaAs/InP SPADs

have different absorption and multiplication materials and there is a larger cut-off wavelength in the narrow band absorption layer. It is also the main reason why the crosstalk probability of InGaAs/InP arrays is much larger than Si arrays^{13,22}.” on Page 9, lines 190-193.

4. It is good to show the absolute crosstalk probability in the array (the probability of a secondary crosstalk pulse generation with a first pulse generation in the array) as well as the suppression ratio, like reference 22 and others.

Response: The reviewer made a good suggestion to show the absolute crosstalk probability, which can reflect the true crosstalk level of the array. However, the absolute crosstalk probability would show a big difference depending on the excess bias voltage, quenching circuits, avalanche duration, etc. In this paper, we focus on showing the crosstalk suppression of the CES. It was evaluated mainly by the DC crosstalk measurements and pseudo-crosstalk measurements². The *emitter* was biased above the breakdown at a constant current ($>10 \mu\text{A}$) to emit crosstalk photon, as a stable crosstalk source. In this case, the $N_{\text{crosstalk}}$ and crosstalk counts obtained are a value proportional to the absolute crosstalk probability. But the crosstalk suppression of the CES obtained by these methods are also reliable. The CES achieves $\sim 90\%$ high crosstalk suppression, as shown in Fig. 3. In addition, we predict that the CES may show better performance when the *emitter* is also operated in pulse counting mode, since the *emitter* will emit much fewer crosstalk photons compared to the constant crosstalk source.

We have added the relevant reference (Rech, I. et al.) to the revised manuscript on Page 18, line 379 (Methods, DC crosstalk measurements); Page 24, lines 490-491 (reference 37).

Fig. 3 The measured crosstalk suppression of the CES between the nearest pixels at 300 K. Measured results **a** when the *detector* was operated in Linear mode; and **b** when the *detector* was operated in Geiger mode. The error bar represents the fluctuation in counts between repeated tests.

5. Figure 4 is a bit busy. Could you consider separating these subfigures into two figures and expanding them?

Response: We have separated Figure 4 into two figures, Figure 4, and Figure 5, as expected; and legends have also been revised on Pages 26-27, lines 537-555.

6. Minor comment

- Subscript characters (e.g. CES in V_CES) are roman font.

Response: Thank you for pointing this out, we have carefully proofed the manuscript and corrected the errors according to the formatting requirements, such as " $N_{\text{crosstalk}}$ ", " V_{ces} ", "DCR", " d " etc.

References

1. Wu, D. R., Tsai, C. M., Huang, Y. H. & Lin, S. D. Crosstalk Between Single-Photon Avalanche Diodes in a 0.18- μm High-Voltage CMOS Process. *J. Lightwave Technol.* **36**, 833-837 (2018).
2. Rech, I. et al. Optical crosstalk in single photon avalanche diode arrays: a new complete model. *Opt. Express* **16**, 8381-8394 (2008).
3. Buttafava, M. et al. SPAD-based asynchronous-readout array detectors for image-scanning microscopy. *Optica* **7**, 755-765 (2020).
4. Calandri, N., Sanzaro, M., Motta, L., Savoia, C. & Tosi, A. Optical Crosstalk in InGaAs/InP SPAD Array: Analysis and Reduction With FIB-Etched Trenches. *IEEE Photon. Technol. Lett.* **28**, 1767-1770 (2016).

REVIEWERS' COMMENTS

Reviewer #1 (Remarks to the Author):

The authors properly replied to all the comments of the reviewers and they significantly improved the manuscript. In my opinion, the manuscript is now suitable for publication on Nature Communications.

Minor comment: There is a typo in Figure 1, where "Charge" is written as "Chagre".

Reviewer #2 (Remarks to the Author):

I appreciate that the authors have provided new measurements and data to satisfy my previous review comments. The new data supports the authors' claim that the DC crosstalk measurement can be used to study the crosstalk suppression in the pulse counting mode. The new data fulfills the weak point of the discussion in the previous version.

Also, the added explanation on Page 9, lines 190-193 makes the content clearer for readers who are not experts on InGaAs/InP arrays to understand the difference between conventional Si-arrays and the InGaAs/InP arrays.

On that basis, I basically agree with the publication in this journal. But, before the publication, let me point out the figures. The presentation of figures should be improved throughout the manuscript. Figures 4 and 5 are separated into two figures, but many figures are still difficult to see. For example, in Figures 4b and 5c, the black lines and text overlapping the blue areas are not well visualized. Consider color assignment. Also, is it indispensable to show the entire 0-50um region on the horizontal axis?

Besides that, the text size of the legend needs to be increased on many of the figures.

Reviewer #1 (Remarks to the Author):

Response: We sincerely appreciate your review and valuable comments on our manuscript.

The authors properly replied to all the comments of the reviewers and they significantly improved the manuscript. In my opinion, the manuscript is now suitable for publication on Nature Communications.

Minor comment: There is a typo in Figure 1, where "Charge" is written as "Chagre".

Response: Thanks to the reviewer for pointing out this error. It has been modified in Figure 1.

Reviewer #2 (Remarks to the Author):

Response: We sincerely appreciate your review and valuable comments on our manuscript.

I appreciate that the authors have provided new measurements and data to satisfy my previous review comments. The new data supports the authors' claim that the DC crosstalk measurement can be used to study the crosstalk suppression in the pulse counting mode. The new data fulfills the weak point of the discussion in the previous version.

Also, the added explanation on Page 9, lines 190-193 makes the content clearer for readers who are not experts on InGaAs/InP arrays to understand the difference between conventional Si-arrays and the InGaAs/InP arrays.

On that basis, I basically agree with the publication in this journal. But, before the publication, let me point out the figures. The presentation of figures should be improved throughout the manuscript. Figures 4 and 5 are separated into two figures, but many figures are still difficult to see. For example, in Figures 4b and 5c, the black lines and text overlapping the blue areas are not well visualized. Consider color assignment. Also, is it indispensable to show the entire 0-50 μ m region on the horizontal axis?

Besides that, the text size of the legend needs to be increased on many of the figures.

Response: Thanks to the reviewer for pointing out this. We have modified the figures in the manuscript to make them easier to see. In Figures 4b and 5c, we have changed the color of the lines and the text, and we also increased the size of lines. The description of the streamlines in the manuscript has also been revised on Page 11, line 227 and Page 27, lines 548, 560.

And to enhance visibility, we display limited simulation areas in Figures 4b, 4c, and 5c, rather than the complete 0-50 μ m simulation regions. It is important to note that this does not alter the discussion of phenomena in the manuscript. The 0-40 μ m region is presented in Figures 4b and 4c, whereas the 0-35 μ m region is depicted in Fig. 5c. And on Page 11, lines 225-227 and Page 27, lines 550, 562-563 of the manuscript, an associated illustration has been added.

The text size of the legend has been increased.